# The Square-Root Unscented and the Square-Root Cubature Kalman Filters on Manifolds

**DOI:** 10.3390/s24206622

**Published:** 2024-10-14

**Authors:** Joachim Clemens, Constantin Wellhausen

**Affiliations:** Cognitive Neuroinformatics Group, University of Bremen, 28359 Bremen, Germany; wellhausen@uni-bremen.de

**Keywords:** unscented Kalman filter, square-root unscented Kalman filter, cubature Kalman filter, square-root cubature Kalman filter, manifolds, state estimation, orientation estimation, localization, autonomous systems

## Abstract

Estimating the state of a system by fusing sensor data is a major prerequisite in many applications. When the state is time-variant, derivatives of the Kalman filter are a popular choice for solving that task. Two variants are the square-root unscented Kalman filter (SRUKF) and the square-root cubature Kalman filter (SCKF). In contrast to the unscented Kalman filter (UKF) and the cubature Kalman filter (CKF), they do not operate on the covariance matrix but on its square root. In this work, we modify the SRUKF and the SCKF for use on manifolds. This is particularly relevant for many state estimation problems when, for example, an orientation is part of a state or a measurement. In contrast to other approaches, our solution is both generic and mathematically coherent. It has the same theoretical complexity as the UKF and CKF on manifolds, but we show that the practical implementation can be faster. Furthermore, it gains the improved numerical properties of the classical SRUKF and SCKF. We compare the SRUKF and the SCKF on manifolds to the UKF and the CKF on manifolds, using the example of odometry estimation for an autonomous car. It is demonstrated that all algorithms have the same localization performance, but our SRUKF and SCKF have lower computational demands.

## 1. Introduction

State estimation is one of the most essential tasks in automated and autonomous systems. It refers to determining a usually time-variant state, which cannot be observed directly, by fusing information from one or multiple sensors. A common example is localization, i.e., estimating the position and altitude of a vehicle or other objects in an environment.

Due to their computational efficiency, variants of the Kalman filter [1] are popular choices for solving state estimation problems. They assume that the uncertainty of all models and the posterior is normally distributed, and they estimate the state in terms of its mean and covariance. For non-linear systems, the extended Kalman filter (EKF) [2], the unscented Kalman filter (UKF) [3], and the cubature Kalman filter (CKF) [4] are three of the most fundamental implementations. The EKF uses a first-order Taylor series expansion in order to linearize the models, while the latter two employ deterministic sampling techniques. The UKF uses the so-called sigma points as samples, and the CKF uses cubature points, which are generated in a slightly different way. In contrast to the EKF, the UKF and the CKF are derivative-free and tend to be more accurate, but they also have a higher computational complexity. An enhancement of the UKF and the CKF is the square-root UKF (SRUKF) [5] and the square-root CKF (SCKF) [4], respectively, which do not track the covariance matrix but tracks its square root. In the case of state estimation, they have the same theoretical complexity as the corresponding non-square-root versions, but the practical implementation can be faster. For the special case of parameter estimation, the algorithms can be modified, resulting in an improved theoretical complexity [5]. Finally, they are numerically more stable.

A challenge in state estimation is the handling of manifold spaces. They behave like a vector space locally but have a more complex global topology. Particularly in the case of localization, one is almost always facing the special orthogonal group SO(n), which is the space of all rotations in *n*-dimensional space (usually n=2,3) and has a circular topological structure. In other words, whenever an orientation in 2D or 3D is part of a state or a measurement, one has to deal with a manifold. This applies to many practically relevant applications, including but not limited to aerospace, ground vehicles, maritime applications, household robotics, and logistics. However, many estimation algorithms, including the basic variants of the Kalman filter, can only operate on vector spaces and cannot be applied here. As a result, dedicated implementations are needed, and a detailed discussion on that can be found in [6,7]. There exist multiple approaches for the EKF [8,9,10,11], the UKF [7,12], and the CKF [13]. However, to the best of our knowledge, there is no generic and coherent implementation of the SRUKF or the SCKF on manifolds.

In this paper, we present modified versions of the SRUKF and the SCKF for use on manifold spaces, which are based on the ⊞-method (pronounced “boxplus-method”) [7]. With those algorithms, one can benefit from the improved properties of a square-root filter in all applications where the state space or the measurement space is a manifold. We make a one-to-one comparison of the SRUKF on manifolds to the SRUKF for vector space and discuss all necessary changes in detail. Furthermore, we show how to convert the SRUKF on manifolds into a SCKF on manifolds. The resulting algorithms can be applied to almost arbitrary manifolds and are not limited to any particular ones. We use the challenging problem of localizing an autonomous vehicle in 3D Euclidean space as an example application in order to demonstrate the effectiveness of our approach. In particular, the proposed SRUKF and SCKF are compared to the UKF and the CKF on a real-world dataset. While maintaining the same localization performance, the SRUKF and the SCKF are shown to be computationally more efficient.

The remainder of the paper is structured as follows. We first review related work regarding, in particular, other attempts to apply the SRUKF and the SCKF to manifolds as well as applications of the ⊞-method. Furthermore, a brief overview of other advances in Kalman-filtering techniques is given. In Section 3, we discuss the basics of encapsulating manifolds using the ⊞-method and corresponding challenges. In Section 4, we recap the classical SRUKF and derive our SRUKF approach for use on manifolds. After that, in Section 5, the necessary changes to turn the SRUKF on manifolds into a SCKF on manifolds are discussed. The example application and a detailed empirical evaluation are presented in Section 6. The paper concludes with a summary and an outlook.

## 2. Related Work

There are some works on using the SRUKF and the SCKF on manifold spaces. But to the best of our knowledge, there is no solution that is both generic and coherent. In [14], an SRUKF is used for visual–inertial navigation, with a quaternion being part of the state space. The parameters of the quaternion are treated as a vector. Accordingly, the state space has more degrees of freedom than the underlying manifold space, and thus, the resulting constraints (unit quaternion) need to be enforced by a workaround. A quaternion is also used in the state vector of [15], where the SRUKF is applied to spacecraft relative navigation. Here, a three-dimensional vector representing the attitude error is used when generating the sigma points, which solves the issue of an over-parametrized state space. While this is conceptually similar to the ⊞-method used in our approach, the particular solution is specific to quaternions. Furthermore, the algorithm cannot deal with measurement spaces that are manifolds. A similar error-state formulation is employed in [16] for using the SCKF in a strap-down inertial navigation system. This also applies to [17], where the SCKF is used for attitude estimation. Accordingly, those approaches have the same disadvantages. A detailed analysis on how to treat quaternions in a UKF, and a possible solution is presented in [18]. The findings are applied to the SRUKF as well and a corresponding algorithm is proposed. Both filters are able to handle not only quaternions in the state but also quaternions as measurements. However, the treatment of the state and the measurement space is specific to quaternions. In [19], a SCKF for attitude estimation is proposed. It also allows for using quaternions in the state and as measurements. But this approach is specific to quaternions as well and thus cannot be applied to other manifolds.

A more general approach to using the UKF and the SRUKF on constraint state spaces is proposed in [20], which also has similarities to the ⊞-method. However, the algorithm does not apply a final transformation of the covariance matrix (or, more precisely, its Cholesky factor), which is required to correctly represent the posterior state uncertainty. This particular problem is discussed more in detail in the context of the UKF in [7] and in Section 3 of this work. Furthermore, the approach also does not support measurements in manifold spaces. Both apply to [21] as well, where the SCKF is modified for use on manifolds by employing Lie groups. It also does not perform the final transformation of the covariance matrix, and the measurements are restricted to vector spaces. In [22], another more general algorithm is presented, which utilizes Lie groups to implement an SRUKF on manifolds. The need for a final transformation of the state uncertainty is addressed here. The proposed solution is to apply a linearization using a first-order Taylor series expansion, which is also discussed in Section 3 (see (5) and (6)). However, this approach is undesirable in a derivative-free filter like the SRUKF. In [23], the PhD thesis corresponding to [18], an SRUKF on Riemannian manifolds is proposed. Since this algorithm is very general, it cannot be embedded within a generic framework, which would allow for a convenient handling of practically relevant manifolds (e.g., SO(n), SE(n), and combinations of those with $R^{n}$). Instead, it requires the definition of manifolds-specific operations in order to apply the approach to a particular problem, which results in a significant implementation effort.

One of the most popular approaches for generic filtering on manifolds is the use of Lie groups [8,9,12,13,21,22,24]. The general idea is to represent the state as a group element, i.e., in a manifold space, while filter operations take place in the tangent space, which is a vector space. The ⊞-method, which is used for our approach, has been proposed in [7]. It is mathematically basically equivalent to filtering on Lie groups but offers a more convenient notation by encapsulating the group operations. The formalism has been used to implement various estimation algorithms on manifolds. In the original publications, modified variants of least squares and the UKF are presented. An implementation of the EKF on manifolds is derived in [10] and further improved in [11]. The latter publication also demonstrates how to realize a particle filter on manifolds using the ⊞-method. Another popular application is graph optimization [25], which is particularly relevant in the context of graph-based simultaneous localization and mapping [26]. Most recently, [27] utilized the ⊞-method for applying the interacting multiple model filter and smoother to manifold spaces.

There are other advances in Kalman-filtering techniques beyond the SRUKF and SCKF that are worth mentioning. However, since we are modifying existing approaches and do not propose a fundamentally new one, we leave the detailed comparison of the different algorithms to the corresponding papers. Furthermore, we are focusing on conceptional developments and do not include application-specific improvements. One advancement is the combination of interacting multi-model techniques with the EKF (IMM-EKF). Here, the filter can switch between different models based on the development of uncertainty in order to adapt to changing conditions. This has, for example, been applied to vehicle localization [28] and multi-target state estimation [29]. The technique has also been applied to the UKF, which is consequently referred to as interacting multi-model UKF (IMM-UKF). Possible applications are, for example, sensor fault detection for unmanned aerial vehicles [30] or sensor calibration for underwater vehicles [31]. Another development is the multi-state constraint Kalman filter (MSCKF) [32], which is based on the EKF. Here, the filter tracks multiple instances of the state at different time points, which is particularly relevant for visual odometry. There is also a cubature-variant of the MSCKF [33]. Furthermore, IMM techniques have most recently been combined with the MSCKF in order to form the IMM-MSCKF [34]. Finally, the invariant EKF (IEKF) [24] is another significant advancement in recent years. It is mainly used for filtering on Lie groups and uses an alternative formulation of the EKF, where the filter gain becomes invariant of the estimated state. As a result, the filter has a better convergence behavior and is more stable in cases of a large deviation between the estimate and the true state. An invariant UKF (IUKF) [12] and invariant CKF (ICKF) [13,35] have been proposed as well. In addition to the Kalman filter, the unscented transform has been applied to the particle filter, which is referred to as the unscented particle filter (UPF) [36]. There exists also a square-root variant, which is the square-root UPF (SRUPF) [37].

## 3. Encapsulation of Manifolds Using the ⊞-Method

The ⊞-method is based on two observations: First, manifolds locally behave like a vector space. Second, many algorithms that may potentially be used on manifolds (examples see previous section) mainly manipulate the elements of the state space using addition and subtraction. This is exploited by encapsulating a manifold state space S using two operators $⊞:S\times R^{n}\mapsto S$ and $⊟:S\times S\mapsto R^{n}$, where *n* denotes the number of degrees of freedom in S. The operator ⊞ adds a small perturbation given in $R^{n}$ to an element in S, which yields an element in S. The operator ⊟ calculates the difference between two elements in S, which yields an element in $R^{n}$. Thus we have
(1)$$\begin{array}{cc} y & =x⊞d\;, \end{array}$$
(2)$$\begin{array}{cc} d & =y⊟x\;,\; \end{array}$$
with x,y∈S and $d\in R^{n}$. A manifold space with both box operators is referred to as a ⊞-manifold. Further relevant properties, mathematical derivations, as well as the definitions of those operators for different manifold spaces can be found in [7].

In order to extend an algorithm defined for vector spaces to work on ⊞-manifolds, one basically needs to replace +/− with ⊞/⊟. However, many fusion algorithms rely on probability distributions in order to represent the uncertainty of the estimate and the sensor measurements. Most commonly, a normal distribution is used. Its definition can be extended to ⊞-manifolds using [7]
(3)$$N_{S}\left(\mu ,\Sigma \right)=\mu ⊞N\left(0,\Sigma \right)\;,$$
with mean μ∈S and covariance $\Sigma \in R^{n\times n}$. Here, N is an ordinary multivariate normal distribution defined on vector spaces, while $N_{S}$ is a normal distribution on S. When clear from the context, the subscript of $N_{S}$ is omitted in the following.

An important implication of (3) is that the mean cannot be changed without updating the covariance as well [7]. In particular, when adding a perturbation *d* to μ using μ⊞d (e.g., in the correction step of a Kalman filter), Σ is still defined with respect to the old mean. This results in an inconsistency since
(4)μ⊞N(d,Σ)≠(μ⊞d)⊞N(0,Σ).

For derivation-based algorithms, like the EKF on ⊞-manifolds [11], the problem can be solved by linearizing μ⊞d using a first-order Taylor series expansion developed around μ and transforming Σ accordingly. This gives us
(5)$$\begin{array}{cc} \mu^{\prime } & =\mu ⊞d\;, \end{array}$$
(6)$$\begin{array}{cc} \Sigma^{\prime } & =J_{d}\Sigma J_{d}^{⊤}\;, \end{array}$$
with $J_{d}=\frac{\partial \mu ⊞d}{\partial \mu }$. However, as discussed already, this approach is not desired in derivation-free algorithms.

Consequently, in the UKF on ⊞-manifolds [7], an additional sigma point transformation is performed to retrieve the updated mean and covariance. In particular, instead of using the ⊞-operator to apply *d* to μ directly, sigma points with a corresponding deviation are generated, which are then used to compute the new mean and covariance. This results in
(7)X=μ⊞dμ⊞(d+Σ)μ¯⊞(d−Σ),(8)μ′=MEANOFSIGMAPOINTS(X),(9)Σ′=12∑i=02nXi⊟μ′Xi⊟μ′⊤,
where MEANOFSIGMAPOINTS is given in Table 1. Further details on this approach can be found in [7] and are discussed in the context of the SRUKF in the following section.

**Table 1 sensors-24-06622-t001:** Method for computing the mean $\mu^{\prime }$ of a set of sigma points Y, where $w_{i}$ corresponds to the weight associated with the *i*-th sigma point. (Algorithm adopted from [7]).

MEANOFSIGMAPOINTS
Input:
(10)Yi,i=0,…,2n
Determine mean $\mu^{\prime }$:
(11)μ0′=Y0(12)μk+1′=μk′⊞∑i=02nwiYi⊟μk′(13)μ′=limk↦∞μk′

## 4. Square-Root Unscented Kalman Filter on Manifolds

In this section, we first review the classical SRUKF for regular vector spaces. After that, we introduce and discuss the necessary modifications in order to apply the SRUKF to ⊞-manifolds. Both algorithms are given in Table 2 next to each other.

### 4.1. Classical Square-Root Unscented Kalman Filter

In the classical SRUKF [5], the state $x_{t}\in R^{n}$, the state transition measurement $u_{t}\in R^{m}$, and the correction measurement $z_{t}\in R^{k}$ are regular vectors, where *t* is the current time step. Furthermore, as for the majority of Kalman filters, the state is assumed to be normally distributed given the set of measurements, i.e.,
(14)$$x_{t}|u_{1:t},z_{1:t}∼N\left(\mu_{t},\Sigma_{t}\right)\;.$$
The filter also estimates the mean $\mu_{t}\in R^{n}$. But instead of determining the covariance $\Sigma_{t}\in R^{n\times n}$, it tracks its square root $S_{t}=\sqrt{\Sigma_{t}}$. The square root is calculated once on initialization using a Cholesky decomposition. After that, the filter operates on the Cholesky factors, which are propagated and updated accordingly.

Thus, the inputs in (15) of one time step of the SRUKF are the previous estimate defined by $\mu_{t-1},S_{t-1}$, the current state transition measurement $u_{t}$, and the current correction measurement $z_{t}$. Furthermore, the uncertainty of the state transition and of the correction measurement are given in terms of covariance matrices $U_{t}\in R^{n\times n},Z_{t}\in R^{k\times k}$ of the corresponding additive, normally distributed noise, where, again, the square-root form is used.

**Table 2 sensors-24-06622-t002:** Comparison of the classical SRUKF for vector spaces and the modified SRUKF for ⊞-manifolds.

Classical SRUKF	SRUKF on ⊞-Manifolds
Input:	Input:
(15)μt−1,St−1,ut,zt,Ut,Zt	(31)μt−1,St−1,ut,zt,Ut,Zt
Prediction Step:	Prediction Step:
(16)Xt−1=μt−1μt−1+ηSt−1μt−1−ηSt−1(17)X¯t*=f(Xt−1,ut)(18)μ¯t=∑i=02nwiX¯t,i*(19)S¯t=qrw1X¯t,1:2n*−μ¯tUt(20)S¯t=cholupdateS¯t,X¯t,0*−μ¯t,w0	(32)Xt−1=μt−1μt−1⊞ηSt−1μt−1⊞−ηSt−1(33)X¯t*=f(Xt−1,ut)(34)μ¯t=MEANOFSIGMAPOINTS(X¯t*)(35)S¯t=qrw1X¯t,1:2n*⊟μ¯tUt(36)S¯t=cholupdateS¯t,X¯t,0*⊟μ¯t,w0
Correction Step:	Correction Step:
(21)X¯t=μ¯tμ¯t+ηS¯tμ¯t−ηS¯t(22)Z¯t=h(X¯t)(23)z^t=∑i=02nwiZ¯t,i(24)Stz=qrw1Z¯t,1:2n−z^tZt(25)Stz=cholupdateStz,Z¯t,0*−z^t,w0(26)Σtx,z=∑i=02nwiX¯t,i−μ¯tZ¯t,i−z^t⊤(27)Kt=(Σtx,z/Stz⊤)/Stz(28)μt=μ¯t+Ktzt−z^t(29)Vt=KtStz(30)St=cholupdateS¯t,Vt,−1	(37)X¯t=μ¯tμ¯t⊞ηS¯tμ¯t⊞−ηS¯t(38)Z¯t=h(X¯t)(39)z^t=MEANOFSIGMAPOINTS(Z¯t)(40)Stz=qrw1Z¯t,1:2n⊟z^tZt(41)Stz=cholupdateStz,Z¯t,0*⊟z^t,w0(42)Σtx,z=∑i=02nwiX¯t,i⊟μ¯tZ¯t,i⊟z^t⊤(43)Kt=(Σtx,z/Stz⊤)/Stz(44)δt=Ktzt⊟z^t(45)Vt=KtStz(46)St′=cholupdateS¯t,Vt,−1(47)Xt′=μ¯t⊞δtμ¯t⊞(δt+ηSt′)μ¯t⊞(δt−ηSt′)(48)μt=MEANOFSIGMAPOINTS(Xt′)(49)St=qrw1Xt,1:2n′⊟μt(50)St=cholupdateSt′,Xt,0′⊟μt,w0

#### 4.1.1. Prediction Step

The mean $\mu_{t-1}$ is propagated in the same way as in the regular UKF by generating a set of sigma points $X_{t-1}$ for the previous estimate, propagating them through the state transition function $f:R^{n}\times R^{m}\mapsto R^{n}$, and retrieving the predicted mean ${\bar{\mu }}_{t}$ using a weighted sum (see (16)–(18)). The only difference is that the covariance $\Sigma_{t-1}$ is already given in its square-root form $S_{t-1}$, and thus, no Cholesky decomposition needs to be applied when generating the sigma points in (16). Note that each sigma point $X_{t-1,i}\in R^{n}$, i=0,…,2n is a vector with the same dimensionality as the state. For the scaling factor η of the sigma points, we choose $\eta =\sqrt{n+\kappa }$ with κ=3−n according to [3]. Consequently, the weights are set to $w_{0}=\frac{\kappa }{n+\kappa }$ and $w_{i}=\frac{1}{2(n+\kappa )}$, i=1,…,2n.

The propagated Cholesky factor ${\bar{S}}_{t}$ is computed via a QR decomposition followed by a Cholesky update (or downdate). The QR decomposition in (19) is applied to the matrix of the weighted differences between the propagated sigma points ${\bar{X}}_{t}^{*}$ and the propagated mean ${\bar{\mu }}_{t}$ compound with the square root of the state transition noise $R_{t}$. We adopt the notation of [5], where qr{A} is used as shorthand for computing the QR decomposition $A^{⊤}=QR$ and returning the transposed upper triangular matrix of *R*. The Cholesky update in (20) is necessary because $w_{0}$ may be negative, in which case the update actually becomes a downdate. Accordingly, the QR decomposition is only applied to the sigma points 1,…,2n, while the 0-th sigma point is incorporated using the Cholesky up- or downdate. We again follow the notation of [5] and use cholupdate{S,v,±w} for updating the Cholesky factor *S* of $P=AA^{⊤}$ such that it corresponds to the Cholesky factor of the rank-1 update $P\pm \sqrt{w}\;vv^{⊤}$. Furthermore, if *v* is a matrix instead of a vector, cholupdate{·} performs consecutive updates for each column of *v*.

#### 4.1.2. Correction Step

In (21)–(26) of the correction step, the differences between the SRUKF and the UKF are the same as in the prediction step. More precisely, generating the sigma points in (21) does not require a Cholesky decomposition since the propagated Cholesky factor ${\bar{S}}_{t}$ is already given. Furthermore, the Cholesky factor $S_{t}^{z}$ of the measurement error covariance is computed using a QR decomposition in (24) followed by a Cholesky up- or downdate in (25). The computation of the expected measurement ${\hat{z}}_{t}$ using the measurement function $h:R^{n}\mapsto R^{k}$ in (22) and a weighted sum in (23) as well as the calculation of the cross-covariance $\Sigma_{t}^{x,z}$ in (26) are unchanged.

Calculating the Kalman gain $K_{t}$ requires an expensive matrix inversion in the UKF. In the SRUKF, it is instead computed in (27) using two nested inverse solutions of $K_{t}\left(S_{t}^{z}{S_{t}^{z}}^{⊤}\right)=\Sigma_{t}^{x,z}$, which can be implemented using efficient back-substitutions. As in [5], we use x=b/A to denote solving Ax=b for *x*. The Kalman gain is used to weigh the innovation $z_{t}-{\hat{z}}_{t}$, which is subsequently added to the propagated mean ${\bar{\mu }}_{t}$ in order to obtain the posterior mean $\mu_{t}$ in (28). This is, in turn, equivalent to the UKF. For computing the posterior Cholesky factor $S_{t}$ in (29) and (30), we utilize a series of Cholesky downdates. They are applied to the predicted Cholesky factor ${\bar{S}}_{t}$, while the columns of $V_{t}=K_{t}S_{t}^{z}$, i.e., the weighed Cholesky factor of the measurement error covariance, are used as arguments.

This concludes the classical SRUKF algorithm. For further details, including a comparison of the complexity to the UKF, the reader is referred to the original publication [5].

### 4.2. Modifying the Square-Root Unscented Kalman Filter

In order to apply the SRUKF to ⊞-manifolds, a few modifications are required. First, the state $x_{t}\in S$, the state transition measurement $u_{t}\in U$, and the correction measurement $z_{t}\in Z$ are allowed to be elements in arbitrary manifold spaces S,U,Z with corresponding box operators. The number of degrees of freedom of those spaces are n,m,k, i.e., the dimensionality of the corresponding covariance matrices and Cholesky factor remains the same. The posterior $p(x_{t}|u_{0:t},z_{0:t})$ is assumed to be normally distributed as in (14), but here we have a normal distribution on a ⊞-manifold defined according to (3). This gives us
(51)$$x_{t}|u_{1:t},z_{1:t}∼N_{S}\left(\mu_{t},\Sigma_{t}\right)\;,$$
with $\mu_{t}\in S$ being an element in the manifold space of the state. Analogously, the state transition noise and the measurement noise are defined by normal distributions on ⊞-manifolds as well. The filter estimates $\mu_{t}$ and the corresponding Cholesky factor $S_{t}=\sqrt{\Sigma_{t}}$. Accordingly, the input in (31) for one time step of the modified SRUKF is the same as for the classical SRUKF, except that $\mu_{t-1},u_{t},z_{t}$ can be an element in manifold spaces instead of being limited to vector spaces.

#### 4.2.1. Prediction Step

In the first step of the prediction in (32), the sigma points $X_{t-1}$ are generated using the mean $\mu_{t-1}$ and the Cholesky factor $S_{t-1}$ of the previous time step without the need to compute a Cholesky decomposition. Since $\mu_{t-1}$ is a manifold, the perturbation vectors are added using the ⊞-operator. Note that the negative perturbation $\mu_{t-1}-S_{t-1}$ turns into $\mu_{t-1}⊞-S_{t-1}$, since the ⊟-operator is not applicable here. Each resulting sigma point $X_{t-1,i}\in S$, i=0,…,2n is an element in the manifold space S of the state. The sigma points are then propagated through the state transition function f:S×U↦S in (33), which takes elements of manifold spaces as arguments and maps them to a manifold space.

For calculating the mean of sigma points, one cannot use a weighted average because of the potentially complex structure of the underlying manifold. Instead, an iterative approach is employed, which is adapted from [7] and shown in Table 1. This algorithm is used to retrieve the propagated mean ${\bar{\mu }}_{t}$ in (34). As in the classical SRUKF, the propagated Cholesky factor ${\bar{S}}_{t}$ is computed using a QR decomposition followed by a Cholesky up- or downdate (see (35) and (36)). However, since the sigma points, as well as the propagated mean, are elements of a manifold space, the ⊟-operator is used to calculate the differences between the sigma points ${\bar{X}}_{t}^{*}$ and the propagated mean ${\bar{\mu }}_{t}$.

#### 4.2.2. Correction Step

The first part of the correction step is modified analogously to the prediction step (see (37)–(43)). In particular, the ⊞-operator is used to generate the sigma points in (37), the measurement function h:S↦Z in (38) is defined on manifold spaces, and the iterative algorithm of Table 1 is applied to compute the expected measurement in (39). Furthermore, the ⊟-operator is utilized to determine the difference between the sigma points and the expected measurement in the QR decomposition in (40) and in the Cholesky up- or downdate in (41). The same applies to the calculation of the cross-covariance in (42), where the ⊟-operator is used as well. The equation for the Kalman gain in (43) remains unchanged.

A major difference is the update of the mean. An ad-hoc approach would be replacing − with ⊟ and + with ⊞ in (28) for calculating the innovation and adding the weighted innovation to the propagated mean. This would yield $\mu_{t}={\bar{\mu }}_{t}⊞K_{t}\left(z_{t}⊟{\hat{z}}_{t}\right)\;.$ However, as discussed in Section 3, this would change the mean, while the covariance (or, in our case, the Cholesky factor) is still defined with respect to the old mean (see (4)). This is also not changed by updating the Cholesky factor by incorporating the weighted Cholesky factor of the most recent measurement using (29) and (30).

To resolve this issue, we adopt the approach of the UKF on ⊞-manifolds of [7] (see also (7)–(9)), which we transfer to the SRUKF. First, we compute the state deviation vector $\delta_{t}\in R^{n}$ in (44), which is given by the innovation $z_{t}⊟{\hat{z}}_{t}$ weighted by the Kalman gain $K_{t}$. The innovation is, in turn, determined using the ⊟-operator since the measurement may be in a manifold space. Then the Cholesky factor is updated in (45) and (46) in the same way as in the classical SRUKF. However, the result $S_{t}^{\prime }$ does not become the posterior Cholesky factor but is used as an intermediate value. In particular, new sigma points $X_{t}^{\prime }\in S$ are generated in (47) by adding $\delta_{t}$ and $S_{t}^{\prime }$ as perturbation using the ⊞-operator. Finally, the posterior mean $\mu_{t}$ and Cholesky factor $S_{t}$ are retrieved from the sigma points. For calculating the mean in (48), the iterative approach of Table 1 is used. The Cholesky factor is computed by incorporating the sigma points 1,…,2n using a QR decomposition in (49), followed by a Cholesky up- or downdate in (50) for the 0-th sigma point. The difference between the sigma points and the posterior mean is again determined using the ⊞-operator.

#### 4.2.3. Computational Complexity

Up to (46), the difference in computational complexity between the ⊞-manifold versions of the UKF and the SRUKF is the same as the difference between the classical UKF and SRUKF. The latter is analyzed in [5], the conclusion of which is that both are in $O(n^{3})$. However, even though the theoretical complexity is in the same class, the implementation of particular steps can be more efficient in the SRUKF compared to the UKF. For example, the generation of sigma points requires a Cholesky decomposition of complexity $O(n^{3}/6)$ in the UKF, while in the SRUKF in (16), (21), (32), and (37) no extra computations are required since the Cholesky factor is already given. Furthermore, the back-substitutions for calculating the Kalman gain in (27) and (43) of the SRUKFs are more efficient than the matrix inverse needed in the UKFs.

The same holds true for the extension in (47)–(50), which is necessary for updating the Cholesky factor correctly in the manifold case. In particular, the generation of sigma points in (47) requires no expensive Cholesky decomposition as in the UKF on ⊞-manifolds. The complexity of the QR decomposition (49) is $O(n^{3})$, while the Cholesky update (50) is in $O(n^{2})$. In the UKF on ⊞-manifolds, those steps correspond to computing the posterior covariance, which is implemented using $\sum_{i=0}^{2n}w_{i}\left(X_{t,i}^{\prime }⊟\mu_{t}\right){\left(X_{t,i}^{\prime }⊟\mu_{t}\right)}^{⊤}$. Assuming that $X_{t,i}^{\prime }⊟\mu_{t}$ is precomputed, the complexity of this step is $O(2n^{2}+2n^{3})$.

In summary, the SRUKF with the modifications for ⊞-manifolds is still in $O(n^{3})$ and thus in the same theoretical complexity class as the UKF on ⊞-manifolds. But individual steps can be implemented more efficiently in practice, which has the potential for a faster execution time. This is investigated empirically in the next section. Furthermore, according to [5], the numerical properties of the SRUKF are improved over the UKF, similar to the square-root Kalman filter [38].

## 5. Square-Root Cubature Kalman Filter on Manifolds

Due to the similarities between the SRUKF and the SCKF, it requires only a few changes to turn the SRUKF on manifolds into a SCKF on manifolds. The main difference is that the SRUKF uses sigma points as sample points, while the SCKF uses so-called cubature points. Those are generated in a slightly different way. Furthermore, the recovery of the parameters of the normal distribution from the sample points needs to be adapted. The particular changes to the SRUKF on manifolds algorithm in Table 2 are as follows:The generation of sample points in (32), (37), and (47) is changed to
(52)Xt−1=μt−1⊞ηSt−1μt−1⊞−ηSt−1,(53)X¯t=μ¯t⊞ηS¯tμ¯t⊞−ηS¯t,(54)Xt′=μ¯t⊞η(δt+St′)μ¯t⊞η(δt−St′),
respectively. Note that there is no sample point anymore that corresponds to the mean. Accordingly, the total number of sample points is now 2n instead of 2n+1. For consistency with the other equations, we change the indexing to i=1,…,2n, i.e., the index i=0 is removed.The scaling factor for generating the cubature point is set to $\eta =\sqrt{n}$.The weights of the sample points in (35), (40), (42), and (49) are changed to $w_{i}=\frac{1}{2n}$, i=1,…,2n.Since there is no 0-th sample point anymore, the sum in (42) becomes $\sum_{i=1}^{2n}$, i.e., its lower bound is changed from 0 to 1.There is no weight anymore that could have a negative value, and all sample points are already considered in the QR decompositions in (35), (40), and (49). Accordingly, the Cholesky up- or downdates in (36), (41), and (50) are skipped. Note that the series of Cholesky downdates in (46) is still required for computing the (intermediate) updated Cholesky factor.

Furthermore, the MEANOFSIGMAPOINTS algorithm in Table 1 needs to be adapted as well to match the changes to the sample points:The weights in (12) are changed to $w_{i}=\frac{1}{2n}$, i=1,…,2n.The sum in (12) becomes $\sum_{i=1}^{2n}$.There is no 0-th sample point anymore that corresponds to the mean and can be used for the initialization in (11). Since the remaining sample points are spread around the mean, choosing any other one would lead to a slow convergence of the limit operation in (13). Accordingly, the initial value is set to
(55)$$\mu_{0}^{\prime }=Y_{1}⊞\frac{1}{2}\left(Y_{n+1}⊟Y_{1}\right)\;.$$The sample points $Y_{1}$ and $Y_{n+1}$ are approximately opposite to each other (with respect to the mean), and thus, the chosen value for $\mu_{0}^{\prime }$ is sufficiently close to the actual mean to alow for a fast convergence.

Regarding the computational complexity, the general findings for the SRUKF on manifolds (see Section 4.2.3) hold true as well. The SCKF on manifolds has a slight speed advantage though, since it uses one sample point less (2n instead of 2n+1) and three Cholesky up- or downdates are skipped. Accordingly, the computational complexity is improved by a constant factor, but the algorithm remains in the same complexity class.

## 6. Evaluation

We compare the proposed SRUKF on manifolds and SCKF on manifolds to the UKF on manifolds [7] and the CKF on manifolds. For the latter, we modify the UKF on manifolds analogously to the changes discussed in Section 5. All filters are implemented in C++ using the Eigen library (https://eigen.tuxfamily.org, accessed on 26 August 2024). For the implementation of the SRUKF and the SCKF, there are a few points to consider: The QR decomposition in Eigen is based on Househoulder transformations. To gain a speed advantage, we implement the QR decomposition using the modified Gram-Schmidt algorithm [39] (Sect. 5.2.8), which is further optimized for computing *R* only, since we are not interested in *Q*. Furthermore, Eigen’s matrix storage order should be taken into account to benefit from CPU caching. The Cholesky update or downdate is computed in-place. One could also convert constant covariance matrices (e.g., for sensor noise) to their square-root forms once in order to save time for the additional computations in each step, but this is not performed here for the commonality with the non-square-root filters.

In this section, we first introduce the example application as well as the corresponding dataset and used parameters. After that, we evaluate the performance of the filters in terms of state estimation accuracy and runtime.

### 6.1. Example Application: Inertial Odometry

We use relative localization (i.e., odometry) of an autonomous car as an example application for our evaluation. For this purpose, the vehicle [40] is equipped with an inertial measurement unit (IMU), wheel odometers, and a sensor measuring the steering wheel angle. The application, all corresponding models, and the use of the UKF on manifolds to solve the estimation problem are described in detail in [41]. The values to be estimated are the 3D position in $R^{3}$, the 3D orientation in SO(3), the 3D velocity in $R^{3}$, as well as the 3D accelerometer and 3D gyroscope biases, each in $R^{3}$. Since SO(3) is a manifold, the joint state space is a manifold as well, and a corresponding filter needs to be used.

One challenge with this particular estimation task is that there is no external source for directly or indirectly correcting the position and the rotation around the global z-axis. This is no issue for the application itself since we are only interested in odometry, i.e., estimating the relative movement of the vehicle. But it results in a growing covariance matrix, which may result in instabilities of the filter. Moreover, the (SR)UKF and the (S)CKF may fail completely, since the generated sample points are not meaningful anymore for very large covariance values.

The solution proposed in [41] is to estimate the uncertainty with respect to a reference state, which is moved forward in time. By that means, the covariance matrix remains within a reasonable range. The moving reference is implemented using two Kalman filters, which process the same measurements but use different references. After a defined period of time or driven distance, the uncertainty of one filter is transferred to the other filter, while the uncertainty of the first filter is set to zero. Again, full details and all derivations can be found in the original publication.

In the UKF and the CKF, the uncertainty transfer and the uncertainty reset are each implemented using a propagation of sample points followed by recovering the mean and covariance. To apply this concept to the SRUKF on manifolds, we employ the familiar modifications: The sigma points are generated using the Cholesky factor without the need for an additional Cholesky factorization, the mean is calculated using the MEANOFSIGMAPOINTS function (see Table 1), and the updated Cholesky factor is retrieved using a QR decomposition followed by a Cholesky up- or downdate. This is, in essence, equivalent to the prediction step (see (32)–(36) in Table 2) except for that the state transition function *f* is replaced by the functions for the uncertainty transfer or the uncertainty reset defined in [41]. Furthermore, no additional noise is introduced, i.e., $U_{t}$ is omitted. For the SCKF, the uncertainty transfer and the uncertainty reset are implemented in the same way as for the SRUKF, but with the changes discussed in Section 5, i.e., we generate cubature points instead of sigma points, the scaling factor and the weights are adapted, and there is no need for a Cholesky up- or downdate to retrieve the updated Cholesky factor. Furthermore, the modified version of MEANOFSIGMAPOINTS is used.

### 6.2. Dataset and Parameters

To the best of our knowledge, there exists no public dataset that contains all desired measurements (IMU, speed, steering angle, and RTK reference system) for our application. For that reason, we use a self-recorded dataset, which consists of four complementary scenarios, which are listed in Table 3. The first scenario takes place on a parking lot with partially harsh maneuvers, including sharp accelerating and decelerating as well as high turning speeds. In the second scenario, we are driving through a suburb of Bremen, Germany, with comparably low speeds. The third scenario is recorded in the inner city of Chemnitz, Germany, and has medium speeds. The fourth and longest scenario also takes place in Chemnitz, Germany, but the majority of the time, we are driving on an urban freeway with higher speeds.

The ground-truth position is given by RTK-corrected GNSS measurements provided by a ublox F9P receiver at 10 Hz. When the GNSS solution has a quality of “RTK fix” (i.e., integer solution), the position accuracy is within 0.05 m. All data points with a worse quality are excluded. For the datasets “Parking lot”, “City”, and “Urban freeway”, the vehicle was equipped with a second ublox F9P receiver operating in moving base mode, which provides a precise heading (and pitch) estimate with an accuracy of less than 1°. In the case of the “Suburb” dataset, a second receiver was not available, and the heading is estimated from the vehicle’s movement. The assumed accuracy is about 5°. The ground-truth trajectories are shown in Figure 1.

All parameters used for the evaluation are listed in Table 4. They are equal for both filters. If possible, the noise parameters for the sensors have been estimated from the data. Otherwise, the values have been set by experts and were tuned for the best performance of the filters.

### 6.3. Relative Trajectory Error

Since the state is estimated without absolute position corrections, the relative trajectory error is the most relevant accuracy measure. As proposed in [42], we use segments of constant lengths (100 m, 200 m, 500 m, 1000 m, and 2000 m), align the first pose estimate of each segment with the corresponding ground-truth pose, and then compute the translation and rotation error for the pose at the end of the segment. The values are normalized with respect to the respective segment length, resulting in the translation error being in percent, while the rotation error is given in degrees per meter. For computing the errors and generating the plots, the script of [43] is used.

The resulting box plots are shown in Figure 2. The median of the translation error is between 7% and 1%, while the rotation error is between 0.04 °/m and 0.005 °/m. For all datasets, the rotation error decreases with the segment length. For the “Parking lot” and the “Urban freeway” scenario (see Figure 2a,d), this also holds true for the translation error, while for “Suburb” and “City” (see Figure 2b,c), it is rather constant. The smallest translation error can be observed for “Parking lot”, which can be explained by the short overall distance. The “Urban freeway” scenario, with its long distance and high speeds, has the highest translation error, particularly for short segment lengths. The “Suburb” dataset has the highest rotation error, which is caused by the fact that a different IMU is used here. Regarding the scope of this work, the most significant insight is that the SRUKF and the SCKF have exactly the same performance as the UKF and the CKF.

### 6.4. Absolute Trajectory Error

Due to the lack of absolute position corrections (see above), the absolute trajectory error is less relevant. It is nevertheless analyzed for completeness. A difficulty is that a small estimation error at the beginning (in particular in the orientation) can have a huge impact on the overall error, even though the remaining trajectory has been estimated with high accuracy. To mitigate this effect, we align the estimated trajectory with the ground-truth using an SE(2) transformation (see [43] for further details). After that, the root-mean-square error (RMSE) of the position is computed over the whole trajectory.

The aligned trajectories are shown in Figure 3 and Figure 4. All estimated trajectories show a small drift compared to the ground-truth, which cannot be avoided without absolute position corrections. Most noticeable is a deviation at the beginning of the “City” scenario (top right in Figure 3c and Figure 4c). This is likely caused by the fact that the estimate of the IMU biases did not match the true values while they converged later. It is again evident that the performance of all filters is equivalent since the estimated trajectories of the UKF and the SRUKF, as well as the trajectories of the CKF and the SCKF, align perfectly with each other.

This is confirmed by the quantitative errors given in Table 5. In case of “Parking lot”, the RMSE of all filters is exactly the same, except for the SCKF, where the RMSE is 1 mm smaller. For “Suburb” the RMSE differs between 3 mm and 7 mm. The biggest difference can be seen in the “City” scenario, where the RMSE of the cubature filters is almost 0.5 m larger than the RMSE of the unscented filters. However, as discussed above, all filters had difficulties estimating the IMU biases correctly at the beginning of that scenario and a minor orientation error has a large impact on the absolute positioning error. Finally, in the “Urban freeway” scenario, the RMSE of the UKF and the CKF are equal, and the RMSE of the SRUKF and the SCKF are equal as well. At the same time, the error of the square-root filters is 8 mm smaller than the error of the non-square-root variants. Note that the errors are calculated over several kilometers, i.e., the deviations between the filters are negligibly small.

### 6.5. Runtime

Finally, we evaluate the runtime of the algorithms on a desktop computer with an Intel® Core™ i7 13700K (16 cores, 5.40 GHz, Intel Corporation, Santa Clara, CA, USA) and 64 GB RAM. Note that the filters do not make use of parallelization and, thus, utilize only one core. The results are averaged over 10 runs.

The total duration for processing each dataset, as well as the percentage difference between the filters, are given in Table 6. Related to the duration of the respective scenarios (see Table 3), all filters are able to process the measurements in real time. The speed advantage of the SRUKF over the UKF is between 3% and 5%, while the speed advantage of the SCKF over the CKF is between 3% and 6%. The CKF is slightly faster than the UKF, with an advantage of up to 3%, while in the “City” scenario, it is 0.5% slower. In comparison to the SRUKF, the SCKF is between 1% and 2% faster. Consequently, the largest difference is between the UKF and the SCKF, where the latter has a speed advantage of 4% to 7%. Overall, this confirms the expectations: The square-root filters are faster than the non-square-root filters, while the cubature filters are, in general, faster than the unscented filters.

## 7. Conclusions

In this work, we modified the SRUKF and the SCKF for use on manifolds. More precisely, not only the state but also the measurements can be elements in almost arbitrary manifold spaces. This is, for example, particularly relevant for state estimation when an orientation is part of a state vector or a measured quantity, where the classical SRUKF and SCKF cannot be applied. We employ the ⊞-method for our approach, which offers a sound way to encapsulate manifold spaces. In contrast to other solutions, our SRUKF on manifolds and our SCKF on manifolds are both generic and mathematically coherent. They have the same theoretical complexity as the UKF on manifolds and the CKF on manifolds, respectively, but the practical implementation can be faster. In addition, as the classical SRUKF and SCKF, they have improved numerical properties compared to the UKF and the CKF. Furthermore, for parameter estimation, the algorithm can be modified for a lower computational complexity. The only disadvantage is that the implementation effort is a little bit greater.

We demonstrated the effectiveness of our approach using the example of relative localization (i.e., odometry) of an autonomous car, where the 3D orientation of the vehicle is part of the state vector, and thus, the state space is a manifold. For this purpose, we used four real-world datasets of different scenarios (parking lot, suburb, city, and urban freeway). It was shown that the SRUKF and the SCKF on manifolds have the same localization performance as the UKF and the CKF on manifolds in terms of relative translation and rotation error as well as the absolute positioning error. At the same time, our SRUKF and our SCKF have lower computational demands than their non-square-root counterparts.

One of our next steps is to compare the SRUKF and the SCKF to the UKF and the CKF for parameter estimation on manifolds. Furthermore, we plan to apply our SRUKF and our SCKF in different applications. One example is extended object tracking, where the state of a dynamic object (e.g., another traffic participant) is estimated. This state consists of a position, orientation, size, and velocity, and thus, it is an element of a manifold [44]. Another possible application is maritime robotics, where we aim to use the proposed approach for localizing an unmanned surface vehicle. The state space is also a manifold here since the 3D orientation is part of the state. Furthermore, angular heading and pitch measurements are incorporated, so we have a manifold measurement space as well.

## Figures and Tables

**Figure 1 sensors-24-06622-f001:**
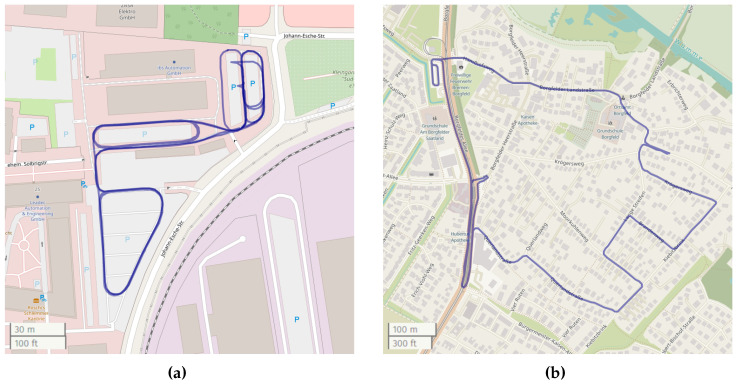
Ground-truth trajectories of the evaluation dataset. (Maps generated using UMap based on OpenStreetMaps data). (**a**) Parking Lot; (**b**) Suburb; (**c**) City; (**d**) Urban freeway.

**Figure 2 sensors-24-06622-f002:**
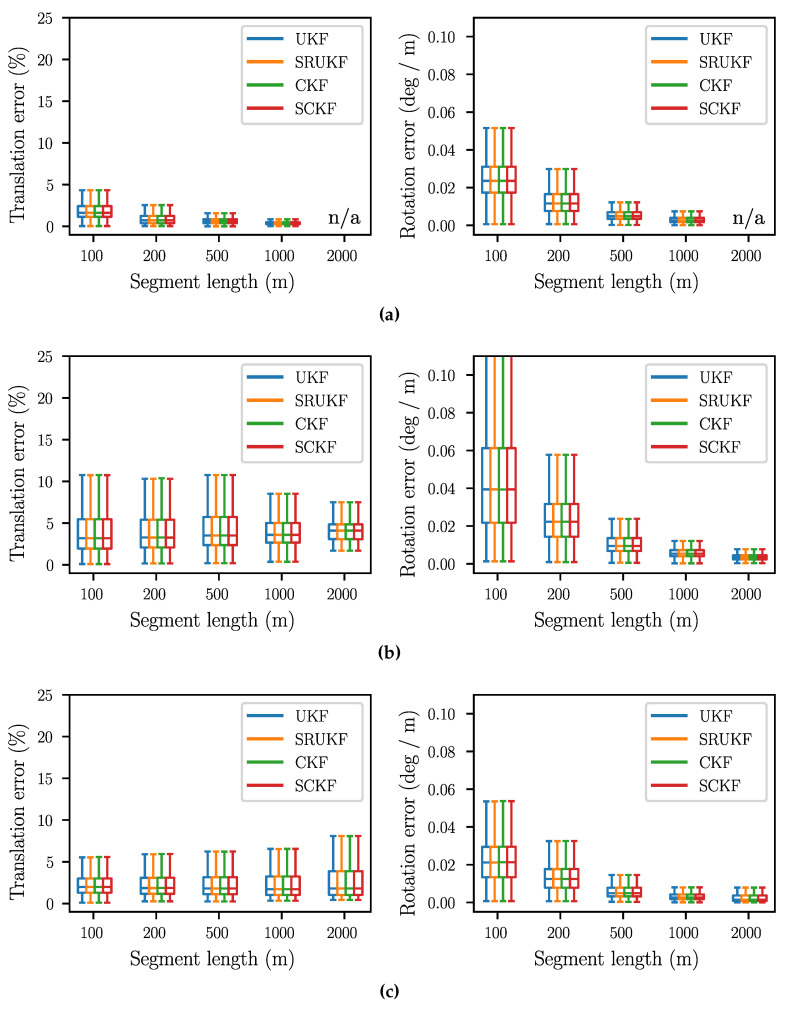
Relative translation and rotation error for different segment lengths. (**a**) Parking Lot; (**b**) Suburb; (**c**) City; (**d**) Urban freeway.

**Figure 3 sensors-24-06622-f003:**
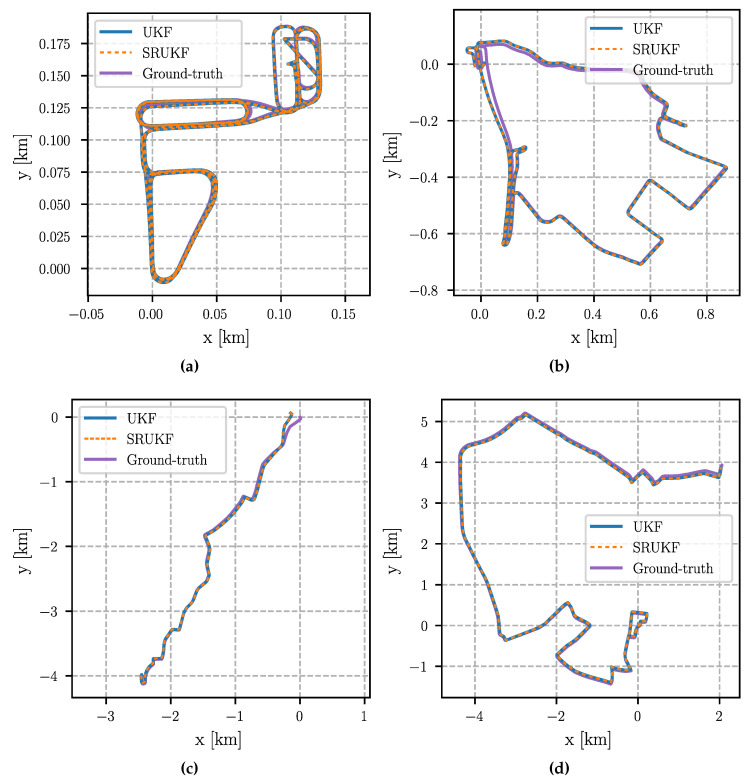
Estimated trajectories of the UKF and SRUKF aligned using an SE(2) transformation to the ground-truth trajectories. Note the different scale of the figures. (**a**) Parking Lot; (**b**) Suburb; (**c**) City; (**d**) Urban freeway.

**Figure 4 sensors-24-06622-f004:**
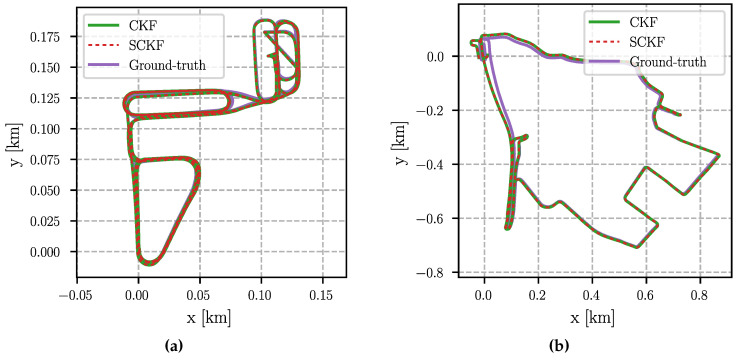
Estimated trajectories of the CKF and SCKF aligned using an SE(2) transformation to the ground-truth trajectories. Note the different scale of the figures. (**a**) Parking Lot; (**b**) Suburb; (**c**) City; (**d**) Urban freeway.

**Table 3 sensors-24-06622-t003:** Datasets.

Name	Duration	Distance	Median Speed	Max. Speed	Trajectory
Parking lot	11:11	3.01 km	17.5 km/h	48.8 km/h	Figure 1a
Suburb	20:46	4.51 km	13.8 km/h	51.0 km/h	Figure 1b
City	13:19	5.52 km	21.1 km/h	60.9 km/h	Figure 1c
Urban freeway	42:28	25.80 km	38.6 km/h	101.6 km/h	Figure 1d

**Table 4 sensors-24-06622-t004:** Parameters (noise given as standard deviation).

Name	Value	Remark
accelerometer noise	0.15 m/s^2^	for each axis
gyroscope noise	0.15 °/s	for each axis
accelerometer bias random walk	0.01 m/s^2^/s	for each axis
gyroscope bias random walk	10^−6^ °/s/ s	for each axis
accelerometer bias init noise	1.0 m/s^2^	for each axis
gyroscope bias init noise	1.0 °/s	for each axis
speed noise	0.1 m/s	
zero speed noise y-axis	0.3 m/s	see [41] for further details
zero speed noise z-axis	0.25 m/s	see [41] for further details
steering angle noise	2.0 °	
covariance reference time	10 s	see [41] for further details
covariance reference distance	0.5 m	see [41] for further details

**Table 5 sensors-24-06622-t005:** Absolute root mean square (RMS) translation error.

Algorithm	Parking Lot	Suburb	City	Urban Freeway
UKF	2.612 m	45.509 m	35.990 m	57.536 m
SRUKF	2.612 m	45.512 m	35.991 m	57.528 m
CKF	2.612 m	45.513 m	36.444 m	57.536 m
SCKF	2.611 m	45.516 m	36.466 m	57.528 m

**Table 6 sensors-24-06622-t006:** Runtime of the algorithms and percentage difference.

Algorithm	Parking Lot	Suburb	City	Urban Freeway
UKF	6.31 s	6.47 s	9.72 s	29.21 s
SRUKF	6.15 s	6.14 s	9.42 s	28.17 s
CKF	6.26 s	6.31 s	9.77 s	28.46 s
SCKF	6.08 s	5.99 s	9.19 s	27.63 s
UKF vs. SRUKF	97.5%	94.9%	96.9%	96.4%
UKF vs. CKF	99.2%	97.5%	100.5%	97.4%
UKF vs. SCKF	96.4%	92.6%	94.5%	94.6%
CKF vs. SRUKF	98.2%	97.3%	96.4%	99.0%
CKF vs. SCKF	97.1%	94.9%	94.1%	97.1%
SRUKF vs. SCKF	98.9%	97.6%	97.6%	98.1%

## Data Availability

The data used in this research are available at https://github.com/JoachimClemens/AD-Datasets (accessed on 25 August 2024).

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
