# Peer review of "The Square-Root Unscented and the Square-Root Cubature Kalman Filters on Manifolds"

_sensors, 2024, doi:10.3390/s24206622_

Round 1
Reviewer 1 Report
Comments and Suggestions for Authors
This paper proposed a square-root unscented Kalman filter on manifolds.
The following comments must be addressed before publication.
1. Pages30-32: "In the case of state estimation it has the same theoretical complexity, but the practical implementation can be faster. For the special case of parameter estimation, the theoretical complexity is improved as well." it is difficult to understand. Please explain it and add citations.
2. It is recommended to add the following citation in the introduction, in which a UKF is combined with DNN to speed up the estimation process.
[1] Zhou X, Qiao D, Li X. Neural Network-Based Method for Orbit Uncertainty Propagation and Estimation[J]. IEEE Transactions on Aerospace and Electronic Systems, 2024, 60(1): 1176-1193.
3. It is recommended to opensource the datasets to ensure the results can be reproduced.
4. In the figures, the results of UKF are overlapped by the results of SRUKF. Please use dashed lines to show the results of the UKF.
Author Response
Please see the attached PDF document.

Reviewer 2 Report
Comments and Suggestions for Authors
This document tries a variant of the square-root unscented Kalman filter, which can be used on state estimation problems where the state space behaves like a manifold. This is a viable research direction. Overall, this draft was organized well. There are still some points that need to be clarified.
- What motivated this research? This should be stated clearly in the introduction. What are the key applications where this technology may be used?
- Is the “box plus” method representation typical in this research direction?
- In Section 4.1, especially on equation 50, what does t represent (time step or instant)?
- It would be helpful to include additional use cases.
It would be helpful to discuss the pros and cons and future directions (e.g. using with machine learning for state estimation).
Author Response
Please see the attached PDF document.

Reviewer 3 Report
Comments and Suggestions for Authors
Please find enclosed.

The language is generally good, but a few sentences need to be checked.
Author Response
Please see the attached PDF document.

Round 2
Reviewer 1 Report
Comments and Suggestions for Authors
thanks for your revisions.